# Calcitriol Protects against Acetaminophen-Induced Hepatotoxicity in Mice

**DOI:** 10.3390/biomedicines11061534

**Published:** 2023-05-25

**Authors:** Supachaya Sriphoosanaphan, Pakkapon Rattanachaisit, Kanjana Somanawat, Natcha Wanpiyarat, Piyawat Komolmit, Duangporn Werawatganon

**Affiliations:** 1Division of Gastroenterology, Department of Medicine, Faculty of Medicine, Bangkok 10330, Thailand; supachaya.sr@gmail.com (S.S.); piyawat.komolmit@gmail.com (P.K.); 2Center of Excellence in Liver Diseases, King Chulalongkorn Memorial Hospital, Thai Red Cross Society, Bangkok 10330, Thailand; 3Center of Excellence in Alternative and Complementary Medicine for Gastrointestinal and Liver Diseases, Department of Physiology, Faculty of Medicine, Chulalongkorn University, Bangkok 10330, Thailand; pakkaoak@gmail.com (P.R.); somanawat55@hotmail.co.th (K.S.); 4Department of Pathology, Faculty of Medicine, King Chulalongkorn Memorial Hospital, Bangkok 10330, Thailand; natchawanpi@gmail.com

**Keywords:** vitamin D, calcitriol, acetaminophen, APAP, hepatotoxicity, liver injury

## Abstract

Acetaminophen (APAP) overdose is one of the major causes of acute liver failure. Severe liver inflammation and the production of oxidative stress occur due to toxic APAP metabolites and glutathione depletion. Growing evidence has proved that vitamin D (VD) exerts anti-inflammatory and antioxidative functions. Our objective was to explore the protective role of calcitriol (VD3) in acute APAP-induced liver injury. **Methods**: Adult male mice were randomized into three groups; control (n = 8), APAP (n = 8), and VD3 group (n = 8). All mice, except controls, received oral administration of APAP (400 mg/kg) and were sacrificed 24 h later. In the VD3 group, calcitriol (10 µg/kg) was injected intraperitoneally 24 h before and after exposure to APAP. Blood samples were collected to assess serum aminotransferase and inflammatory cytokines [tumor necrosis factor-alpha (TNF-α) and interleukin-6 (IL-6)]. Liver tissues were analyzed for hepatic glutathione (GSH), malondialdehyde (MDA), and histopathology. **Results**: APAP administration significantly increased serum aminotransferase, inflammatory cytokines, and induced cellular inflammation and necrosis. APAP also depleted hepatic GSH and elevated oxidative stress, as indicated by high MDA levels. In the APAP group, 25% of the mice (two out of eight) died, while no deaths occurred in the VD3 group. Treatment with calcitriol significantly reduced serum aminotransferase, TNF-α, and IL-6 levels in the VD3 group compared to the APAP group. Additionally, VD3 effectively restored GSH reserves, reduced lipid peroxidation, and attenuated hepatotoxicity. **Conclusions**: These findings demonstrate that VD3 prevents APAP-induced acute liver injury and reduces mortality in mice through its anti-inflammatory and antioxidative activity. Thus, VD3 might be a novel treatment strategy for APAP-induced hepatotoxicity.

## 1. Introduction

Acetaminophen (N-acetyl-para-aminophenol; APAP) is widely used as an analgesic and antipyretic drug. Despite its relatively safety, APAP overdose remains a significant cause of acute liver injury and acute liver failure worldwide [1,2]. APAP-induced liver injury is characterized by a marked elevation in serum hepatic aminotransferase, hepatocellular degeneration, and hemorrhagic centrilobular necrosis [3]. At the therapeutic level, APAP is metabolized to nontoxic metabolites in the liver via the glucuronidation and sulfation pathways. Unfortunately, excessive intake of APAP upregulates these pathways and enhances the formation of a reactive toxic metabolite called N-acetyl-p-benzoquinone imine (NAPQI) through cytochrome P450 (CYP) enzymes [4,5]. NAPQI is efficiently detoxified by hepatic glutathione (GSH) and removed from cells. However, when GSH levels become depleted, unconjugated NAPQI binds to cellular proteins, resulting in the formation of APAP-protein adducts. These protein adducts play a crucial role in initiating mitochondrial dysfunction, which leads to an excess production of oxidative stress. The reactive oxygen species (ROS) provokes lipid peroxidation, DNA fragmentation, hepatic inflammation, and cellular necrosis, exacerbating the liver injury caused by APAP [4,5,6,7].

The main standard treatment for APAP overdose is N-acetyl-cysteine (NAC), which serves as a precursor of GSH. NAC helps restore intracellular GSH levels, facilitate the excretion of NAPQI, and reduce excess ROS [8]. Nonetheless, NAC carries a risk of anaphylactoid reactions and is generally more effective when administered to patients presenting early after APAP overdose [9]. Due to these limitations, there is a need for novel alternative treatment options. Several compounds with anti-inflammatory and antioxidant properties have shown potential in protecting against APAP injury in animal models. However, only a limited number of these compounds are currently investigated in human clinical studies [10,11,12].

Vitamin D (VD) is an essential hormone responsible for calcium homeostasis and bone metabolism. Apart from its classic function, emerging experimental evidence has highlighted the anti-inflammatory and antioxidant properties of VD in various conditions including liver diseases [13,14,15]. VD exerts its effects by binding to the VD receptor (VDR), which is abundantly expressed in hepatocytes and immune cells [16]. Upon binding to VDR, VD regulates intracellular signaling pathways involved in numerous physiological processes. Importantly, during liver inflammation, the VDR in hepatocytes has been observed to be upregulated, expanding the potential targets for VD action [17]. In an animal model of nonalcoholic fatty liver disease (NAFLD), a VD analog exhibited promising results by reducing liver inflammation and hepatic steatosis [18]. Additionally, VD deficiency was found to exacerbate hepatic oxidative stress and inflammation in mice with APAP-induced liver injury, suggesting a potential role of VD in mitigating APAP-induced hepatotoxicity [19]. 

To date, there is a lack of comprehensive data regarding the role of VD in APAP-induced hepatoxicity. Only a limited number of studies conducted in a rat model have explored the potential protective effects of VD supplementation against APAP-related liver injury [20,21]. However, it should be noted that rat models may not accurately reflect the pathophysiology observed in humans due to differences in the severity of liver injury and the type of cellular necrosis [22]. On the contrary, mouse models closely resemble the pathophysiological changes observed in humans. Therefore, the objective of this study is to ascertain whether VD3 can mitigate APAP-induced hepatoxicity using a mouse model. Specifically, our objective is to assess changes in liver injury, inflammatory markers, lipid peroxidation, hepatic GSH levels, and liver histopathology to determine the potential therapeutic effects of VD in this context.

## 2. Materials and Methods

### 2.1. Animals

Adult male ICR mice (*Mus musculus*; age: 4–5 weeks, body weight: 25–30 g) were purchased from the National Laboratory Animal Center, Mahidol University, Thailand. The animals were acclimatized at least 1 week in a 12 h light-dark cycle in controlled temperature and humidity environment and fed ad libitum until experimental use. The study protocol was approved by the Institutional Animal Care and Use Committee (IACUC), Faculty of Medicine, Chulalongkorn University (IRB No.013/2566). The study was carried out according to the Ethical Principles and Guidelines for the Use of Animals by the National Research Council of Thailand (1999).

### 2.2. Paracetamol and VD3 Preparation

Acetaminophen (Tylenol^®^) was dissolved in freshly prepared distilled water for the experiment. A single dose of 400 mg/kg of APAP was administered to mice by oral gavage. An active form of VD, calcitriol (VD3) was used for an intervention in this study. The dose of APAP was chosen based on our previous studies [10,11], in which we observed significant biochemical changes and histological injury in animals following APAP administration. 

Calcitriol injection formula (Cacare^®^, Nang Kuang pharmaceutical company, Tapei, Tawan) was injected intraperitoneally into mice in the VD3-treated group. A dose of 10 µg/kg was administered at two intervals, both before and after APAP exposure.

### 2.3. Experimental Protocol

Twenty-four mice were randomly assigned into three experimental groups (n = 8, each); Control group, APAP group, and VD3 group. After 12 h fasting period, the mice in the APAP group and the VD3 group were fed a 400 mg/kg single dose of APAP orally via an intragastric tube. In the VD3 group, VD3 was injected intraperitoneally at two time points: 24 h before and immediately after APAP administration. The mice were sacrificed at 24 h later following APAP exposure. 

All mice were anesthetized by the intraperitoneal injection of thiopental (50 mg/kg). A medial incision was made to expose the liver and then the entire liver was rapidly removed and washed with cold normal saline (4–8 °C). Liver samples were collected, immediately frozen in liquid nitrogen, and stored at −80 °C for the measurement of hepatic GSH and MDA levels. The remaining liver tissue was fixed in a 10% formalin solution for histopathological evaluation. Whole blood was withdrawn from heart, and the serum was obtained by allowing the blood to coagulate at room temperature for 2 h, followed by centrifugation at 1000× *g* for 20 min at 4 °C. Serum samples were collected to assess the level of transaminase [serum glutamic oxaloacetic transaminase (SGOT) and serum glutamate-pyruvate transaminase (SGPT)] and inflammatory cytokines [interleukin-6 (IL-6) and tumor necrosis factor-alpha (TNF-α)].

### 2.4. Serum Hepatic Aminotransferase Levels

Serum SGOT and SGPT levels were analyzed with Reflotron^®^ system (Roche diagnostics, Mannheim, Germany) following the manufacturer’s instructions. The final level was reported as U/L.

### 2.5. Serum IL-6 Immunoassay

The serum level of IL-6 was measured with a quantitative sandwich enzyme-linked immunosorbent assay (ELISA) technique according to the manufacturer’s instructions (Quantikine^®^ ELISA, R&D Systems, Minneapolis, MN, USA). The results were calculated by reference to the standard curve. The IL-6 concentration was reported as pg/mL.

### 2.6. Serum TNF-α Immunoassay

Serum concentration of TNF-α was measured via the quantitative sandwich ELISA technique following the protocols by the manufacturer (Quantikine^®^ ELISA, R&D Systems, Minneapolis, MN, USA). The results were calculated by referencing the standard curve. The final concentration was reported as pg/mL.

### 2.7. Hepatic GSH Assay

Hepatic GSH level was measured with the glutathione assay kit (Cayman Chemical Company, Ann Arbor, MI, USA) according to the manufacturer’s instruction. Liver tissues were dissected and homogenized in denaturing lysis buffer (pH 7.4) and centrifuged at 10,000× *g* for 15 min at 4 °C. The GSH content was determined via a standard enzymatic recycling procedure. The sulfhydryl group of glutathione (GSH) reacted with 5,5′-ditrio-bis-(2-nitrobenzoic acid) (DTNB), forming a yellow compound called 5-thio-2-nitrobenzoic acid (TNB). The mixed disulfide GSTNB, formed between GSH and TNB, was then reduced by GSH reductase, facilitating GSH recycling and additional TNB generation. The rate of TNB production was directly proportional to the hepatic GSH concentration. GSH levels in the sample were accurately estimated by quantifying TNB’s optical density (O.D.) at 405–414 nm using a microplate reader. The GSH content was expressed as nmol/mg protein.

### 2.8. Hepatic MDA Assay

Hepatic MDA level, a naturally occurring product of lipid peroxidation, was determined using the Thiobarbituric acid reactive substance (TBARS) assay kit (Cayman Chemical Company, Ann Arbor, MI, USA) following the manufacturer’s instruction. Liver tissues were dissected and homogenized in denaturing lysis buffer and centrifuged at 1600× *g* for 10 min at 4 °C. The 10% trichloroacetic acid (TCA) solution and 1.5 mL of 0.8% Thiobarbituric acid (TBA) solution were added to tissue homogenates. The mixture was boiled in a water bath at 95 °C for 60 min and then cooled with water at room temperature. After centrifugation at 1600× *g* for 15 min, the absorbance of the sample was measured at 530–540 nm. The MDA level was reported as nmol/mg protein.

### 2.9. Histopathology

The liver samples that were fixed in 10% formalin solution were processed by standard preparation for histological specimen. The tissues were embedded in paraffin, sectioned at 5 µm, and stained with Hematoxylin and Eosin (H&E). The histopathological evaluation was performed under a light microscope (LM) by a skilled pathologist who was blinded to the experiment. The hepatic necroinflammation was assessed in each section and a severity grade was given according to the criteria described by Brunt et al. [23] from grade 0 to 3 as follows; score 0 = normal histology, score 1 = sparse or mild focal zone 3 hepatocyte injury/inflammation, score 2 = noticeable zone 3 hepatocyte injury/inflammation, and score 3 = severe zone 3 hepatocyte injury/inflammation.

### 2.10. Statistical Analysis

Data were expressed as mean ± standard deviation (SD). For comparison between two groups or among three groups, an unpaired Student *t*-test and one-way ANOVA with a Student–Newman–Keuls test was used. Data without Gaussian distribution, Kruskal–Wallis with post hoc Mann–Whitney were performed. *p*-value < 0.05 was considered statistically significant. All statistical analyses were performed using the SPSS software (version 22.0; IBM Corp., Amonk, NY, USA) and the graphs were prepared in GraphPad Prism 8 (GraphPad Software, San Diego, CA, USA).

## 3. Results

Among 24 mice, 2 (2/8) mice in the APAP group were dead within 24 h after exposure to APAP. However, the 24 h survival rate was 100% (8/8) in the VD3 group after APAP administration. Therefore, available serum samples and liver tissue were obtained from 22 mice.

### 3.1. Body Weight, Liver Weight, and Liver Index of Mice

Mice in the control group had the lowest liver weight and liver index, while those in the APAP group had the highest indices. The liver index of the mice in the APAP group was significantly higher compared to the control group (7.05 ± 0.22 vs. 5.41% ± 0.07%, *p* < 0.001). On the contrary, mice in the VD3 group had a significantly lower liver index than the APAP group (5.91 ± 0.12% vs. 7.05 ± 0.22%, *p* < 0.001). An organ index for the liver is shown in Appendix A. 

### 3.2. Effects of Calcitriol on Serum Aminotransferase

Serum SGOT levels were significantly elevated in the APAP group compared to the control group (6194.17 ± 4145.41 U/L vs. 180.5 ± 45.71 U/L, *p* = 0.008). In the VD3 group, SGOT significantly decreased after APAP administration when compared to the APAP group (242.13 ± 119.19 U/L vs. 6194.17 ± 4145.41 U/L, *p* = 0.007). Notably, there were no significant differences in serum SGOT levels between the VD3 and the control groups (*p* = 0.085). (Figure 1A).

Similarly, serum SGPT levels were significantly higher in the APAP group compared to control mice (5260.00 ± 4126.01 U/L vs. 24.98 ± 10.12 U/L, *p* = 0.027). Serum ALT levels were significantly lower in the VD3 group than in the APAP group (87.43 ± 61.91 U/L vs. 5260.00 ± 4126.01 U/L, *p* = 0.036). There were no significant differences in serum SGPT levels between the VD3 group and the control group (*p* = 0.085). (Figure 1B).

### 3.3. Effects of Calcitriol on Inflammatory Cytokines

The data of serum inflammatory markers are shown in Figure 1C,D. Serum IL-6 levels in the APAP group were significantly higher than those in the control and the VD3 group [3023.64 ± 222.33 pg/mL vs. 122.77 ± 19.71 pg/mL (*p* < 0.001) and 3023.64 ± 222.33 pg/mL vs. 129.55 ± 55.57 pg/mL (*p* < 0.001), respectively]. There were no significant differences in IL-6 levels between the VD3 group and the control group (*p* = 0.750).

Additionally, serum TNF-α levels were significantly higher in the APAP group compared to the control group (101.41 ± 15.86 pg/mL vs. 1.80 ± 1.11 pg/mL, *p* < 0.001). In VD3-treated mice, serum IL-6 levels were significantly lower than those in the APAP group (5.82 ± 1.98 pg/mL vs. 101.41 ± 15.86 pg/mL, *p* < 0.001).

### 3.4. Effects of Calcitriol on Hepatic GSH

Hepatic GSH levels were significantly decreased in the APAP group compared to those in the control group (3.89 ± 0.37 nmol/mg protein vs. 7.41 ± 0.67 nmol/mg protein, *p* < 0.001). However, hepatic GSH levels were significantly restored in the VD3-treated mice compared to the APAP group (5.92 ± 0.78 nmol/mg protein vs. 3.89 ± 0.37 nmol/mg protein, *p* < 0.001). (Figure 1E).

### 3.5. Effects of Calcitriol on Hepatic Malondialdehyde (MDA)

As shown in Figure 1F, hepatic MDA content was significantly higher in the APAP group compared to both the control and the VD3 group [4.53 ± 1.48 nmol/mg protein vs. 0.36 ± 0.18 nmol/mg protein, (*p* = 0.013) and 4.53 ± 1.48 nmol/mg protein vs. 0.33 ± 0.30 nmol/mg protein (*p* = 0.025), respectively]. Interestingly, there were no significant differences in hepatic MDA concentration between the VD3 group and the control group (*p* = 0.562).

### 3.6. Effects of Calcitriol on Histopathology

The normal appearance and preserved architecture were observed in the control group (Figure 2A,B), whereas there was acute centrilobular hemorrhagic hepatic necrosis and inflammation were observed in all zones in the APAP group (Figure 2C,D). In the VD3 group, alleviation of histological findings was observed compared to the APAP group. Only mild focal inflammation was present, and a significant portion of the hepatic parenchyma remained intact (Figure 2E,F). The mean necroinflammatory score improved in the VD3 group compared to the APAP group (1.00 ± 0.53 vs. 2.67 ± 0.52, *p* < 0.05). A summary of the histopathological findings in the control and all experiment groups is summarized in Table 1.

## 4. Discussion

In the present study, we investigated the effect of VD3 on APAP-mediated hepatotoxicity in a mouse model. Our findings demonstrate that treatment with VD3 resulted in significant reductions in serum hepatic aminotransferase (SGOT and SGPT) and inflammatory cytokines (IL-6 and TNF-α) compared to the APAP group. Furthermore, VD3 administration led to a remarkable attenuation of hepatic GSH and MDA levels, indicating a reduction in oxidative stress and lipid peroxidation in the liver. Histopathological examination also showed improvements in hepatocyte injury following VD3 administration. In addition, VD3 treatment significantly reduced the number of deaths from APAP-induced hepatotoxicity. These findings suggest that VD3 has the potential to attenuate liver injury and improve survival in mice with acute liver injury induced by APAP.

The majority of APAP is mainly metabolized by glucuronidation and sulfation to nontoxic metabolite at therapeutic doses [7]. However, APAP overdose limited these enzymatic activities. As a result, excess APAP is metabolized by CYP P450, particularly CYP 2E1, into NAPQI, which results in depleted GSH reserves in hepatocyte and intracellular protein adduct formation [4,5,22]. These protein adducts triggered mitochondrial oxidative stress and activate c-jun N-terminal kinase (JNK), further exacerbating the oxidative stress. Ultimately, these injuries lead to hepatocyte necrosis and intracellular protein release, which triggers the transcription of pro-inflammatory cytokines such as TNF-α, IL-1β, IL-6, and IL-10 [24,25,26]. These inflammatory mediators finally recruit immune cells, especially neutrophils and monocytes, to the liver, initiating a sterile inflammation response [27,28].

Our findings are in line with previous reports on rat models. The study by El-Boshy et al. showed that VD significantly alleviated hepatorenal damage induced by APAP toxicity [21]. Furthermore, Abood et al., demonstrated that paricalcitol, a VD receptor agonist, had a protective effect against APAP-induced liver injury [20]. However, it should be noted that the pathophysiology of APAP-induced hepatitis can vary between species. Although rats can develop certain aspects of APAP toxicity, the severity is less pronounced compared to mice and may result in milder inflammation and liver injury. On the contrary, mouse models have been widely used and validated to study the molecular cascades involved in APAP toxicity, which closely resemble the pathophysiology observed in humans [4,29,30]. Therefore, our findings in mice ascertained the protective role of VD against APAP-mediated hepatotoxicity.

The role of VD as an immunomodulatory and antioxidative hormone has been extensively studied in both experimental and clinical settings. VD regulates gene transcription via VDR, which is expressed not only in immune cells but also in hepatocytes [31]. VD down-regulated the production of pro-inflammatory cytokines, including TNF-α, IL-6 and IL-8 [32,33]. Moreover, VD has been shown to decrease the production of ROS and inhibit NF-κB signaling, which is a major pathway involved in inflammatory responses [34]. In the clinical stage, VD attenuates liver inflammation and hepatic fibrosis in patients with chronic hepatitis C [35,36]. Additionally, VD acts as an antioxidant to reduce liver injury in patients with autoimmune hepatitis [37].

Understanding the specific mechanism involved is crucial to elucidating the protective effects of VD on APAP-induced hepatotoxicity. The injury in this condition is mediated through various pathophysiological mechanisms. Firstly, the activation of pro-inflammatory cytokines due to extensive damage and death of hepatocytes plays a pivotal role in APAP-induced liver inflammation [13,26,38]. Vitamin D, known for its anti-inflammatory properties, may modulate these inflammatory responses, thereby protecting the liver against further damage. Secondly, APAP-induced hepatic GSH depletion leads to increased oxidative stress and mitochondrial damage, which contribute to liver injury. VD may attenuate this oxidative stress via its antioxidant effect and helps restore the activity of antioxidant enzymes such as glutathione peroxidase, catalase, and superoxide dismutase [21]. Lastly, vitamin D has been found to regulate various signaling pathways associated with cell survival, apoptosis, and the promotion of the regeneration process. These mechanisms may contribute to the hepatoprotective effects of vitamin D in cases of APAP-induced hepatotoxicity [39].

Although there are no specific studies directly examining the dose of VD for APAP-induced hepatotoxicity treatment in a mouse model, it is valuable to refer to the research that demonstrates the anti-inflammatory and antioxidant properties of VD. In particular, a pharmacokinetic study by Muindi et al., provided information on the safe use of VD3 in mice. Ref. [40] The study revealed that intraperitoneal administration of VD3 at a dose of up to 37.5 µg/kg was well tolerated. Given the established anti-inflammatory and antioxidant properties of VD3, as well as its safety profile, we have chosen a dose of 10 µg/kg administered at two intervals (prior and after APAP exposure) to assess the therapeutic efficacy of VD3 in this experiment.

The timing of the VD intervention is one of the important factors determining its therapeutic effect. Previous experimental studies have shown differences in outcomes between pre- and post-treatment protocols of VD supplementation. The prophylactic VD regimen has been reported to yield better results in terms of anti-inflammatory activity [21,41]. By administering VD before the insult occurs, it allows the liver tissue to activate and enhance its antioxidant and anti-inflammatory mechanisms in preparation for the subsequent injury. 

Our study has limitations. The lack of assessment for hepatic CYP2E1, NAPQI, and phase-I and phase-II enzymes limited our understanding of the complete cellular cascades involved. Furthermore, the use of a fixed dose of VD3 precluded the evaluation of the dose–response relationship in APAP-induced hepatotoxicity. Future research should investigate different VD3 doses and measure relevant enzymes and cellular cascades to gain a deeper understanding of its protective mechanism.

In conclusion, we provide evidence that the active form of VD3, calcitriol, protects significantly against APAP-induced hepatotoxicity. VD3 reduced the mortality of APAP-induced liver injury in a mouse model. These beneficial effects are attributed to the attenuation of hepatic inflammation, reduction in oxidative stress, and restoration of hepatic GSH levels during acute APAP toxicity. Based on these findings, VD3 might potentially be a novel therapeutic approach for the treatment of APAP overdose. 

## Figures and Tables

**Figure 1 biomedicines-11-01534-f001:**
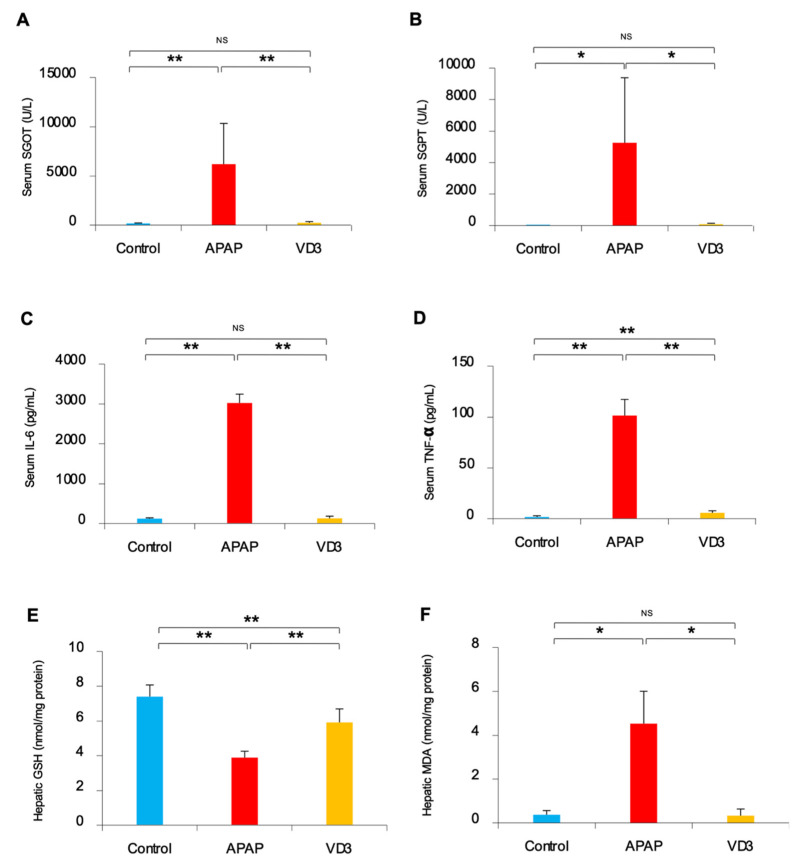
Effects of calcitriol on serum and hepatic biological parameters in mice with APAP-induced hepatitis (**A**): serum glutamic oxaloacetic transaminase (SGOT); (**B**): serum glutamate-pyruvate transaminase (SGPT); (**C**): interleukin-6 (IL-6); (**D**): tumor necrosis factor-α (TNF-α); (**E**): hepatic glutathione (GSH); (**F**): hepatic malondialdehyde (MDA). Horizontal lines indicate the mean and SD, * *p*-value < 0.05, ** *p*-value < 0.01, NS non-significant.

**Figure 2 biomedicines-11-01534-f002:**
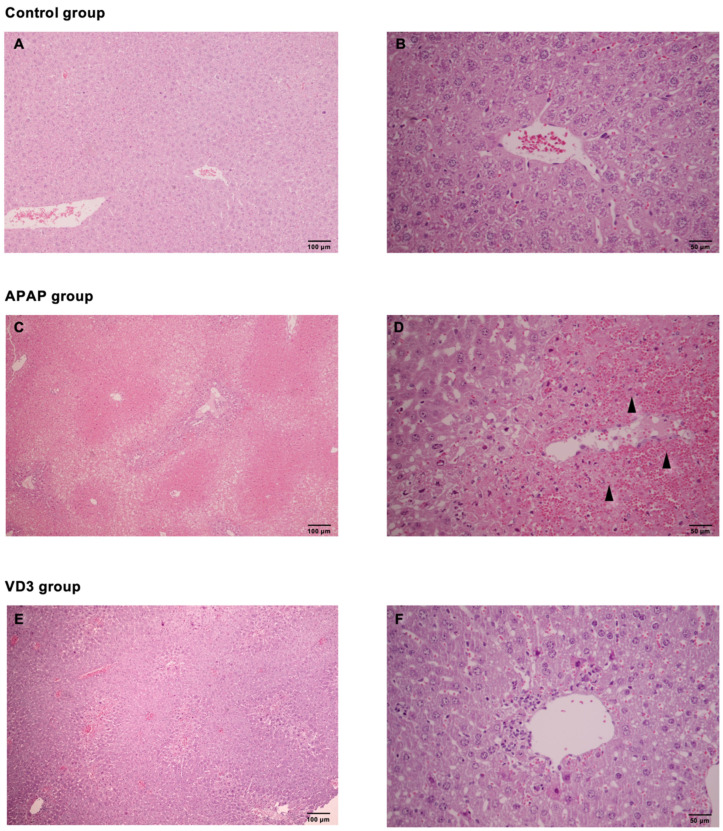
Calcitriol improved liver histopathology in acetaminophen overdose-induced hepatitis mice (Hematoxylin and Eosin—(**A**,**C**,**E**): ×10, (**B**,**D**,**F**): ×40); (**A**,**B**): Control group showed normal liver parenchyma; (**C**,**D**): APAP group showed massive hepatic necrosis with marked inflammation (Arrows indicate the area of hepatic necrosis.); (**E**,**F**): VD3-treated group showed mild hepatic inflammation.

**Table 1 biomedicines-11-01534-t001:** Hepatic necroinflammation grading in all experimental groups.

Group	n	Hepatic Necroinflammation Score				Mean Score	*p*-Value
		0	1	2	3		
Control	8	8	0	0	0	0	<0.001
APAP	6	0	0	2	4	2.67	
VD3-treated	8	1	6	1	0	1.00	

Data present as number of mice exhibited each score of hepatic inflammation. Severity grading criteria was defined by Brunt et al. [23] Comparison was analyzed using Kruskal–Wallis H test with post hoc analysis. APAP; N-acetyl-P-aminophenol, VD3; calcitriol.

## Data Availability

The datasets generated during and/or analyzed during the current study are available from the corresponding author on reasonable request.

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
