# Peer review of "Calcitriol Protects against Acetaminophen-Induced Hepatotoxicity in Mice"

_biomedicines, 2023, doi:10.3390/biomedicines11061534_

Round 1

Reviewer 1 Report

In this work, Sriphoosanaphan at al. studied the protective effect of Calcitriol against acetaminophen-induced hepatotoxicity in mice. They showed that VD3 prevents APAP-induced acute liver injury and reduces mortality in mice through its anti-inflammatory and anti-oxidative activity.

The work has no clear rationale and development. Also, the in vivo drug treatment and related results have severe flaws, as explained below. Moreover, the introduction must be improved according to the present study hypothesis. Please see below:

  1. How did the author fix the dose for VD3 and APAP?
  2. Why author did not measure Phase- I enzymes (cytochrome P450, cytochrome b5, NADPH cytochrome c (P450) reductase and NADPH cytochrome b5 ) and Phase-II enzymes (GST and UDP-GT)
  3. Why did the author use one female mouse for this study?
  4. In figure1, Y axis labels are confusing and need to correct it 
  5. The author should provide an organ index for the liver and spleen.
  6. The author should add hepatic inflammation related makers analysis.

Extensive editing of English language required

Author Response

Reviewer 1

In this work, Sriphoosanaphan et al. studied the protective effect of Calcitriol against acetaminophen-induced hepatotoxicity in mice. They showed that VD3 prevents APAP-induced acute liver injury and reduces mortality in mice through its anti-inflammatory and anti-oxidative activity.

The work has no clear rationale and development. Also, the in vivo drug treatment and related results have severe flaws, as explained below. Moreover, the introduction must be improved according to the present study hypothesis. Please see below:

  1. How did the author fix the dose for VD3 and APAP?

Response: We would like to thank the reviewer for such a thoughtful remark. We agree that varying doses of VD3 and APAP can yield different outcomes. In this experiment, the dose of APAP (400 mg/kg, single dose) was chosen based on our previous studies1,2, in which we observed significant biochemical changes and histological injury in animals following APAP administration. Regarding calcitriol dosage, there may be no specific studies directly examining the dose of VD for APAP-induced hepatotoxicity treatment in a mouse model, it is valuable to reference from the research demonstrating the anti-inflammatory and antioxidant properties of VD. Previous research has investigated the anti-inflammatory and antioxidant properties of VD using varying doses, durations, and administration routes.3-6 The pharmacokinetic study conducted by Muindi et al. provides important insights into the safe use of calcitriol in mice.7 The findings demonstrated that a dose up to 0.75 µg of calcitriol per mouse (equivalent to 37.5 µg/kg) via intraperitoneal administration was well-tolerated. Considering the established anti-inflammatory and antioxidant properties of calcitriol, as well as its safety profile at the aforementioned dose, we have chosen a dose of 10 µg/kg injected at two-time points to evaluate the therapeutic effect of calcitriol in the APAP-induced hepatotoxicity mouse model.

We have included this information to further clarify and enhance the quality of our manuscript.

  1. Why author did not measure Phase- I enzymes (cytochrome P450, cytochrome b5, NADPH cytochrome c (P450) reductase and NADPH cytochrome b5 ) and Phase-II enzymes (GST and UDP-GT)
    Response: We thoroughly agree with the reviewer that incorporating these measurements would provide a more comprehensive understanding of the underlying mechanisms. However, due to limitations in sample availability, we were unable to include the measurement of Phase-I and Phase-II enzymes in our research. We recognize the significance of this limitation and have revised the discussion section accordingly to highlight this important issue. We believe that future investigations should take into consideration the measurement of Phase-I and Phase-II enzymes to further enhance our understanding of calcitriol's protective mechanism.

  1. Why did the author use one female mouse for this study?
    Response: We apologise if there was any confusion caused by the manuscript's wording. We would like to clarify that there was no use of a female mouse in our research. In fact, all the animals involved in the study were male.

  1. In figure1, Y axis labels are confusing and need to correct it
    Response: Thank you for your comment. We do apologise for creating confusion. We have amended Figure 1 as the reviewer suggested.

  1. The author should provide an organ index for the liver and spleen.
    Response: Thank you for raising this issue. We collected each animal's liver and body weight. We agree that the liver index is an important indicator of liver injury and the changes is able to reflect the damage of the liver. In our study, we found that mice in the APAP group had a significantly higher liver index compared to the control group, indicating severe inflammation and organ swelling. While the mice in the VD3 group had significantly lower liver index compared to the APAP group. Moreover, the level in the VD3 group was very close to that of the control group. These findings further supported the effectiveness of calcitriol in mitigating the injury caused by APAP-induced hepatotoxicity.

We have included the data in the result part and Supplementary Table 1.

  1. The author should add hepatic inflammation related makers analysis.
    We appreciate your keen observation and thoughtful suggestion. We thoroughly agree that hepatic inflammation plays a critical role in the progression of APAP hepatotoxicity. Various cytokines, chemokines, and markers have been employed to assess inflammatory status in this context.8,9 Among these markers, TNF-α and IL-6 are commonly analysed to reflect hepatic inflammation in liver diseases, including APAP-induced liver injury.8-11

Given the relevance and significance of TNF-α and IL-6 as indicators of hepatic inflammatory status, we have made the decision to select these markers as outcome for our experiment.

Reference

  1. Somanawat K, Thong-Ngam D, Klaikeaw N. Curcumin attenuated paracetamol overdose induced hepatitis. World J Gastroenterol 2013;19(12):1962-7. DOI: 10.3748/wjg.v19.i12.1962.
  2. Werawatganon D, Linlawan S, Thanapirom K, et al. Aloe vera attenuated liver injury in mice with acetaminophen-induced hepatitis. BMC Complement Altern Med 2014;14:229. DOI: 10.1186/1472-6882-14-229.
  3. Chow EC, Magomedova L, Quach HP, et al. Vitamin D receptor activation down-regulates the small heterodimer partner and increases CYP7A1 to lower cholesterol. Gastroenterology 2014;146(4):1048-59. DOI: 10.1053/j.gastro.2013.12.027.
  4. Jahn D, Dorbath D, Kircher S, et al. Beneficial Effects of Vitamin D Treatment in an Obese Mouse Model of Non-Alcoholic Steatohepatitis. Nutrients 2019;11(1). DOI: 10.3390/nu11010077.
  5. Bozic M, Guzman C, Benet M, et al. Hepatocyte vitamin D receptor regulates lipid metabolism and mediates experimental diet-induced steatosis. J Hepatol 2016;65(4):748-757. DOI: 10.1016/j.jhep.2016.05.031.
  6. Kheder R, Hobkirk J, Saeed Z, et al. Vitamin D(3) supplementation of a high fat high sugar diet ameliorates prediabetic phenotype in female LDLR(-/-) and LDLR(+/+) mice. Immun Inflamm Dis 2017;5(2):151-162. DOI: 10.1002/iid3.154.
  7. Muindi JR, Modzelewski RA, Peng Y, Trump DL, Johnson CS. Pharmacokinetics of 1alpha,25-dihydroxyvitamin D3 in normal mice after systemic exposure to effective and safe antitumor doses. Oncology 2004;66(1):62-6. DOI: 10.1159/000076336.
  8. Lee HC, Liao CC, Day YJ, Liou JT, Li AH, Liu FC. IL-17 deficiency attenuates acetaminophen-induced hepatotoxicity in mice. Toxicol Lett 2018;292:20-30. DOI: 10.1016/j.toxlet.2018.04.021.
  9. Feng Y, Cui R, Li Z, et al. Methane Alleviates Acetaminophen-Induced Liver Injury by Inhibiting Inflammation, Oxidative Stress, Endoplasmic Reticulum Stress, and Apoptosis through the Nrf2/HO-1/NQO1 Signaling Pathway. Oxid Med Cell Longev 2019;2019:7067619. DOI: 10.1155/2019/7067619.
  10. Alkharfy KM, Al-Daghri NM, Yakout SM, Ahmed M. Calcitriol attenuates weight-related systemic inflammation and ultrastructural changes in the liver in a rodent model. Basic Clin Pharmacol Toxicol 2013;112(1):42-9. DOI: 10.1111/j.1742-7843.2012.00936.x.
  11. Gardner CR, Laskin JD, Dambach DM, et al. Exaggerated hepatotoxicity of acetaminophen in mice lacking tumor necrosis factor receptor-1. Potential role of inflammatory mediators. Toxicol Appl Pharmacol 2003;192(2):119-30. DOI: 10.1016/s0041-008x(03)00273-4.

Reviewer 2 Report

Title: Calcitriol protects against acetaminophen-induced hepatotoxicity in mice

In the manuscript, the authors investigated the protective effects of Calcitriol on the APAP-induced mice. In general, the authors have completed a reasonable study with partly informative data on the alleviating effects of Calcitriol. However, the statistical analysis and graphic presentation in the Figures have not been completed in the labeling of the differences on the groups. Moreover, the study on the differences of Calcitriol dosages was not presented for strengthening the choice of the correct dosage.

Specific comments:
1. Twenty-four mice were randomly…

The species of mice, age and body weight should be presented.

2. Positive control
The positive control should be used or discussed in the study for comparison.

3. Figure 2

The scale in the figures should be indicated.

4. Why the edition is in 2021?

5. The line number in each page of the manuscript should be shown. 

Title: Calcitriol protects against acetaminophen-induced hepatotoxicity in mice

In the manuscript, the authors investigated the protective effects of Calcitriol on the APAP-induced mice. In general, the authors have completed a reasonable study with partly informative data on the alleviating effects of Calcitriol. However, the statistical analysis and graphic presentation in the Figures have not been completed in the labeling of the differences on the groups. Moreover, the study on the differences of Calcitriol dosages was not presented for strengthening the choice of the correct dosage.

Specific comments:
1. Twenty-four mice were randomly…

The species of mice, age and body weight should be presented.

2. Positive control
The positive control should be used or discussed in the study for comparison.

3. Figure 2

The scale in the figures should be indicated.

4. Why the edition is in 2021?

5. The line number in each page of the manuscript should be shown. 

Author Response

Reviewer 2

In the manuscript, the authors investigated the protective effects of Calcitriol on the APAP-induced mice. In general, the authors have completed a reasonable study with partly informative data on the alleviating effects of Calcitriol. However, the statistical analysis and graphic presentation in the Figures have not been completed in the labeling of the differences on the groups. Moreover, the study on the differences of Calcitriol dosages was not presented for strengthening the choice of the correct dosage.

Specific comments:

  1. Twenty-four mice were randomly…The species of mice, age and body weight should be presented.
    Response: We apologise that the information on the animal model was unclear. We have revised the manuscript in “Materials and Methods” as follows:

    Adult male ICR mice (Mus musculus; age: 4-5 weeks, body weight: 25-30 g) were purchased from the National Laboratory Animal Center, Mahidol University, Thailand.

  2. Positive control- The positive control should be used or discussed in the study for comparison.
    Response: Thank you for your remark. In our study, we included an APAP group of mice as a positive control to induce hepatocellular injury. Through serum biochemical tests and histopathological assessment, we observed the typical features of hepatocellular injury of hepatocellular injury of APAP-induced hepatotoxicity in this group. Furthermore, in the intervention group receiving VD3, we observed that these changes were mitigated. We have made the amendment to Figure 1 and added p-values to facilitate the comparison between groups to provide a clearer understanding of the results.

  3. Figure 2 - The scale in the figures should be indicated.
    Response: Thank you for your valuable comment. We have revised the pathological figures with the scale per your suggestion.

  1. Why the edition is in 2021?
    Response: Thank you for your comment. We have amended the manuscript edition in the header to 2023.

  2. The line number in each page of the manuscript should be shown. ++
    Response: Thank you for the reviewer’s insightful remark. We have added line numbers throughout the revised manuscript to enhance clarity and facilitate easier referencing.

Reviewer 3 Report

In the submitted manuscript, the hepatoprotective effect of calcitriol on paracetamol-induced liver toxicity was investigated. The mechanisms of the damaging effect of paracetamol on the liver have long been well studied. The significant role of reduced glutathione in the detoxification of the toxic metabolite of paracetamol has been established. Restoring deficient glutathione levels with the administration of NAC is one of the possible detoxification pathways applied in clinical practice. Unfortunately, there are not many alternatives in antidote therapy for acute paracetamol intoxication. Therefore, many scientific groups are working in the field of experimental pharmacology and toxicology, looking for alternatives to NAC. So far, there is no convincing evidence for an alternative treatment for this hepatotoxicity.

The submitted manuscript has some shortcomings that I would like to comment on and accordingly I have some questions.

1. What is the original contribution in this experiment?  You yourself cite data from other studies on the hepatoprotective effect of calcitriol on paracetamol-induced hepatotoxicity.

2. How did you choose the dose and route of administration of vit D? Why only administer one dose? Thus, a dose response relationship cannot be assessed.

3. You have no positive control to compare the effect of vit. D.

4. The mechanism of hepatoprotective action of vit. D in paracetamol-induced hepatotoxicity needs to be well elucidated, bearing in mind the mechanism of paracetamol toxicity. How exactly does vitamin D protect the liver?

5. By what method was the activity of transaminases measured?

6. The possibility of vitamin D-induced hepatotoxicity has not been discussed, given that it is a fat-soluble vitamin. 

English can be improved, minor corrections have to be done.

Author Response

Reviewer 3

In the submitted manuscript, the hepatoprotective effect of calcitriol on paracetamol-induced liver toxicity was investigated. The mechanisms of the damaging effect of paracetamol on the liver have long been well studied. The significant role of reduced glutathione in the detoxification of the toxic metabolite of paracetamol has been established. Restoring deficient glutathione levels with the administration of NAC is one of the possible detoxification pathways applied in clinical practice. Unfortunately, there are not many alternatives in antidote therapy for acute paracetamol intoxication. Therefore, many scientific groups are working in the field of experimental pharmacology and toxicology, looking for alternatives to NAC. So far, there is no convincing evidence for an alternative treatment for this hepatotoxicity.

The submitted manuscript has some shortcomings that I would like to comment on and accordingly I have some questions.

  1. What is the original contribution in this experiment?  You yourself cite data from other studies on the hepatoprotective effect of calcitriol on paracetamol-induced hepatotoxicity.
    Response: Thank you for your insightful comment. Previous studies, which we mentioned in the manuscript, have used rat models to investigate the role of vitamin D in APAP-induced hepatotoxicity. However, it is noteworthy that mouse models have demonstrated similarities to human pathophysiology, showing features such as oncotic necrosis, mitochondrial injury, and nuclear fragmentation in APAP-induced liver injury.1 Furthermore, Wang et al. reported that vitamin D deficiency exacerbated degree of liver injury, hepatic inflammation, and oxidative stress in a mouse model2, providing evidence for the potential role of vitamin D in mitigating APAP-induced hepatotoxicity. As many studies have shown that vitamin D exerts anti-inflammatory and anti-oxidant properties3-5, we aim to further investigate and elucidate the potential benefits of vitamin D in alleviating APAP-induced hepatotoxicity using a mouse model, which closely resembles human pathophysiology.

We have also provided further clarification and elaboration on the rationale of our study in our revised manuscript.

  1. How did you choose the dose and route of administration of vit D? Why only administer one dose? Thus, a dose response relationship cannot be assessed.
    Response: We appreciate your valuable comment. Considering the rapid and severe nature of APAP-induced hepatotoxicity, it is crucial to consider therapeutic interventions that can act promptly. In this regard, we chose to use calcitriol, an active hormonal form of vitamin D. Unlike ergocalciferol and cholecalciferol, calcitriol does not require any further metabolism to exert its effects. Furthermore, a pharmacokinetic study conducted in a mouse model demonstrated the rapid absorption of calcitriol following intraperitoneal administration, without any observed lag phase.6 This pharmacokinetic profile further supported our decision to employ intraperitoneal calciferol in our experiment.

    While there may be no specific studies directly examining the dose of vitamin D for APAP-induced hepatotoxicity treatment in a mouse model, it is valuable to reference from the research demonstrating the anti-inflammatory and antioxidant properties of vitamin D. Varying doses, durations, and administration routes have been explored in previous studies to evaluate the effects of vitamin D in inflammation and oxidative stress.7-10 The pharmacokinetic study conducted by Muindi et al. provides important insights into the safe use of calcitriol in mice.6 The findings demonstrated that a single dose up to 0.75 µg of calcitriol per mouse (equivalent to 37.5 µg/kg) via intraperitoneal administration was well-tolerated. Considering the established anti-inflammatory and antioxidant properties of calcitriol, as well as its safety profile at the aforementioned dose, we have chosen a dose of 10 µg/kg injected at two-time points to evaluate the therapeutic effect of calcitriol in the APAP-induced hepatotoxicity mouse model.

    With a fixed dose of calcitriol used in our experiment, we were unable to assess the dose- response relationship in the context of APAP-induced hepatotoxicity. We recognize the significance of this limitation and have made revisions to the discussion section of our manuscript to emphasize this important aspect.

  2. You have no positive control to compare the effect of vit. D.
    Response: Thank you for your comment. In our study, we included an APAP group of mice as a positive control to induce hepatocellular injury. Through serum biochemical tests and histopathological assessment, we observed the typical features of hepatocellular injury of APAP-induced hepatotoxicity in this group. Furthermore, in the intervention group receiving VD3, we observed that these changes were mitigated. We believed the significant changes between the APAP and VD3 groups reflected the effect of calcitriol. We have also made the amendment to Figure 1 and added p-values to facilitate the comparison between groups to provide a clearer understanding of the results.

  3. The mechanism of hepatoprotective action of vit. D in paracetamol-induced hepatotoxicity needs to be well elucidated, bearing in mind the mechanism of paracetamol toxicity. How exactly does vitamin D protect the liver?
    Response: Thank you for raising an important point. We agree that understanding the specific mechanisms involved is crucial for elucidating the protective effects of vitamin D in this context, particularly considering the mechanism of paracetamol toxicity.

APAP-induced hepatotoxicity is mediated through various pathophysiological mechanisms. First, the activation of pro-inflammatory cytokines and chemokines resulting from massive hepatocyte injury and death is a key component of APAP-induced liver inflammation.4,11,12 Vitamin D, which is known to possess anti-inflammatory property, may modulate these inflammatory responses, thereby protecting the liver against further damage. Second, APAP-induced hepatic GSH depletion leads to increased oxidative stress and mitochondrial damage, which contribute to liver injury. Vitamin D may attenuate this oxidative stress via its anti-oxidant effect and helps restore the activity of anti-oxidant enzymes such as glutathione peroxidase, catalase, and superoxide dismutase.13 Last, vitamin D has been reported to regulate various signalling pathways involved in cell survival, apoptosis, and promote the regeneration process, which could contribute to its hepatoprotective effects in APAP-induced hepatotoxicity.14

While these proposed mechanisms provide insights into the potential hepatoprotective effects of vitamin D, further research using molecular and cellular approaches is necessary to fully elucidate the precise mechanisms involved. We have also amended the discussion part in the revised manuscript to enhance the clarity and improve overall understanding.

  1. By what method was the activity of transaminases measured?
    Response: We appreciate the reviewer’s insightful comment. We have added the measurement of transaminases levels in “Materials and Methods” as follows:

    Hepatic transaminase levels
    SGOT and SGPT were analyzed with Reflotron® system (Roche diagnostics, Mannheim, Germany) following the manufacturer’s instructions. The final level was reported as U/L.

  2. The possibility of vitamin D-induced hepatotoxicity has not been discussed, given that it is a fat-soluble vitamin.
    Response: We thank you for bringing up this important consideration. We completely agree that as a fat-soluble vitamin, vitamin D can accumulate in the liver. However, vitamin D toxicity is uncommon especially in short-term intervention. Previous studies investigating chronic vitamin D overload in murine models have primarily reported biochemical changes in serum calcium rather than hepatotoxicity. Based on a pharmacokinetic study conducted by Muindi et al.6, we believed that calcitriol in our protocol was within the safe range and did not induce hepatotoxicity.

Reference:

  1. Jaeschke H, Xie Y, McGill MR. Acetaminophen-induced Liver Injury: from Animal Models to Humans. J Clin Transl Hepatol 2014;2(3):153-61. DOI: 10.14218/JCTH.2014.00014.
  2. Wang YQ, Geng XP, Wang MW, et al. Vitamin D deficiency exacerbates hepatic oxidative stress and inflammation during acetaminophen-induced acute liver injury in mice. Int Immunopharmacol 2021;97:107716. DOI: 10.1016/j.intimp.2021.107716.
  3. Dauletbaev N, Herscovitch K, Das M, et al. Down-regulation of IL-8 by high-dose vitamin D is specific to hyperinflammatory macrophages and involves mechanisms beyond up-regulation of DUSP1. Br J Pharmacol 2015;172(19):4757-71. DOI: 10.1111/bph.13249.
  4. George N, Kumar TP, Antony S, Jayanarayanan S, Paulose CS. Effect of vitamin D3 in reducing metabolic and oxidative stress in the liver of streptozotocin-induced diabetic rats. Br J Nutr 2012;108(8):1410-8. DOI: 10.1017/S0007114511006830.
  5. Tohari AM, Zhou X, Shu X. Protection against oxidative stress by vitamin D in cone cells. Cell Biochem Funct 2016;34(2):82-94. DOI: 10.1002/cbf.3167.
  6. Muindi JR, Modzelewski RA, Peng Y, Trump DL, Johnson CS. Pharmacokinetics of 1alpha,25-dihydroxyvitamin D3 in normal mice after systemic exposure to effective and safe antitumor doses. Oncology 2004;66(1):62-6. DOI: 10.1159/000076336.
  7. Chow EC, Magomedova L, Quach HP, et al. Vitamin D receptor activation down-regulates the small heterodimer partner and increases CYP7A1 to lower cholesterol. Gastroenterology 2014;146(4):1048-59. DOI: 10.1053/j.gastro.2013.12.027.
  8. Jahn D, Dorbath D, Kircher S, et al. Beneficial Effects of Vitamin D Treatment in an Obese Mouse Model of Non-Alcoholic Steatohepatitis. Nutrients 2019;11(1). DOI: 10.3390/nu11010077.
  9. Bozic M, Guzman C, Benet M, et al. Hepatocyte vitamin D receptor regulates lipid metabolism and mediates experimental diet-induced steatosis. J Hepatol 2016;65(4):748-757. DOI: 10.1016/j.jhep.2016.05.031.
  10. Kheder R, Hobkirk J, Saeed Z, et al. Vitamin D(3) supplementation of a high fat high sugar diet ameliorates prediabetic phenotype in female LDLR(-/-) and LDLR(+/+) mice. Immun Inflamm Dis 2017;5(2):151-162. DOI: 10.1002/iid3.154.
  11. Gardner CR, Laskin JD, Dambach DM, et al. Exaggerated hepatotoxicity of acetaminophen in mice lacking tumor necrosis factor receptor-1. Potential role of inflammatory mediators. Toxicol Appl Pharmacol 2003;192(2):119-30. DOI: 10.1016/s0041-008x(03)00273-4.
  12. Lee HC, Liao CC, Day YJ, Liou JT, Li AH, Liu FC. IL-17 deficiency attenuates acetaminophen-induced hepatotoxicity in mice. Toxicol Lett 2018;292:20-30. DOI: 10.1016/j.toxlet.2018.04.021.
  13. El-Boshy M, BaSalamah MA, Ahmad J, et al. Vitamin D protects against oxidative stress, inflammation and hepatorenal damage induced by acute paracetamol toxicity in rat. Free Radic Biol Med 2019;141:310-321. DOI: 10.1016/j.freeradbiomed.2019.06.030.
  14. El-Sharkawy A, Malki A. Vitamin D Signaling in Inflammation and Cancer: Molecular Mechanisms and Therapeutic Implications. Molecules 2020;25(14). DOI: 10.3390/molecules25143219.

Round 2

Reviewer 1 Report

The authors have satisfactorily responded to all comments and made the necessary changes to the manuscript.

Minor editing of English language required

Author Response

Reviewer 1

Comments and Suggestions for Authors

The authors have satisfactorily responded to all comments and made the necessary changes to the manuscript.

Response: We sincerely appreciate your review and constructive feedback. Your insights have been invaluable in improving our manuscript.

Comments on the Quality of English Language

Minor editing of English language required

Response: We have conducted English polishing as requested.

Reviewer 2 Report

Title: Calcitriol protects against acetaminophen-induced hepatotoxicity in mice

The re-submitted manuscript has been revised according to the reviewer’s comments. Most of the revised contents have been highlighted in red to show the revised text. However, there were some added texts unnecessarily presented in the manuscript. For example, the method in Lines 188-197 should be briefly described. Lines 128-138 were not the methods description but the results. It should be deleted and place in the result or discussion section. Moreover, please rephrase the content in Lines 404-416 for into the concise presentation. Some of the remaining parts would be asked with simplified expression, especially in the method section with the cited reference for showing the experimental procedures.   

Title: Calcitriol protects against acetaminophen-induced hepatotoxicity in mice

The re-submitted manuscript has been revised according to the reviewer’s comments. Most of the revised contents have been highlighted in red to show the revised text. However, there were some added texts unnecessarily presented in the manuscript. For example, the method in Lines 188-197 should be briefly described. Lines 128-138 were not the methods description but the results. It should be deleted and place in the result or discussion section. Moreover, please rephrase the content in Lines 404-416 for into the concise presentation. Some of the remaining parts would be asked with simplified expression, especially in the method section with the cited reference for showing the experimental procedures.   

Author Response

Reviewer 2

Comments and Suggestions for Authors

The re-submitted manuscript has been revised according to the reviewer’s comments. Most of the revised contents have been highlighted in red to show the revised text. However, there were some added texts unnecessarily presented in the manuscript.

For example, the method in Lines 188-197 should be briefly described.

Response: Thank you for your comment. We briefly described hepatic GSH assay in Method.

Lines 128-138 were not the methods description but the results. It should be deleted and place in the result or discussion section.

Response: We acknowledge and agree with the reviewer's suggestion to relocate the paragraph from lines 128-138 to the discussion section.

Moreover, please rephrase the content in Lines 404-416 for into the concise presentation. Some of the remaining parts would be asked with simplified expression, especially in the method section with the cited reference for showing the experimental procedures.   

Response: Thank you for your valuable comment. We have rephrased and simplified the content accordingly.

Reviewer 3 Report

A positive control is a substance or drug with which we compare the hepatoprotective effects of calcitriol, e.g. acetylcysteine, silymarin, etc. Paracetamol is the toxic agent or negative control.

Author Response

Reviewer 3

Comments and Suggestions for Authors

A positive control is a substance or drug with which we compare the hepatoprotective effects of calcitriol, e.g. acetylcysteine, silymarin, etc. Paracetamol is the toxic agent or negative control.

Response: Thank you for your valuable comment. The inclusion of the positive control results is crucial in enhancing our comprehensive understanding of the underlying mechanisms. We appreciate this suggestion and will incorporate it into our future studies.
